# The Role of the CPT Family in Cancer: Searching for New Therapeutic Strategies

**DOI:** 10.3390/biology13110892

**Published:** 2024-11-01

**Authors:** Yanxia Duan, Jiaxin Liu, Ailin Li, Chang Liu, Guang Shu, Gang Yin

**Affiliations:** 1Department of Pathology, Xiangya Hospital, School of Basic Medical Sciences, Central South University, Changsha 410000, China; 226511079@csu.edu.cn (Y.D.); ljx1995@csu.edu.cn (J.L.); liailin0516@outlook.com (A.L.); 2School of Basic Medical Sciences, Central South University, Changsha 410000, China; yylc0916@sina.com; 3National Clinical Research Center for Geriatric Disorders, Xiangya Hospital, School of Basic Medical Sciences, Central South University, Changsha 410000, China; 4China-Africa Research Center of Infectious Diseases, School of Basic Medical Sciences, Central South University, Changsha 410000, China

**Keywords:** CPT1, CPT2, FAO, cancer, oxidative phosphorylation, energy metabolic reprogramming

## Abstract

Numerous studies have demonstrated the key rate-limiting enzymes of the FAO (fatty acid oxidation) pathway, represented by the CPT family, play important roles in tumor proliferation, migration, invasion, stemness, apoptosis, chemoresistance, and metabolic stress. However, the current research on the CPT family focuses more on CPT1A and CPT1B, while relatively few studies have been conducted on CPT1C and CPT2 in tumors. Therefore, we look at molecular structural properties of the CPT family, the roles they play in tumorigenesis and development, the target drugs, and the possible regulatory roles of CPTs in energy metabolism reprogramming to help understand the current state of CPT family research and to search for new therapeutic strategies.

## 1. Introduction

There is increasing evidence that there is substantial metabolic heterogeneity from tumor cells to normal cells, from different tumors, and even from various stages of tumor progression, and that this heterogeneity is associated with patient prognosis, epigenetic status, and treatment resistance [1,2]. Metabolic reprogramming of tumor cells is generally closely linked to disturbances in the balance of glucose and glutamine supply and demand, and it enhances the ability of tumor cells to keep growing and invading other tissue locations [3]. There is a “Warburg effect” on tumor cells, feeding themselves through the inefficient glycolytic pathway even when there is sufficient oxygen [4]. Along with abnormalities in glucose metabolism, disturbances in the balance of lipid catabolism and synthesis have emerged as a new area of cancer metabolism that needs to be studied in depth [5]. Disturbances in lipid metabolic homeostasis, represented by fatty acid oxidation (FAO) imbalance, leading to activation of pro-cancer signals and abnormalities in the expression and activity of related metabolically critical rate-limiting enzymes, have become an important part of metabolic remodeling in cancer [6]. Lipid-metabolizing enzymes, represented by the CPT family, have been reported to act as gatekeepers for the FAO metabolic pathway by playing a key limiting enzyme role in the FAO process in the body, and have been implicated in the activation or inhibition of FAO in various cancers [7]. As PCa (prostate cancer) cells accumulate more lipids through a combination of metabolic changes in the absence of oxygen, this includes reducing fatty acid oxidation, thus protecting the cancer cells from oxidative stress and endoplasmic reticulum stress [8]. Targeted inhibition of CPT1A can reduce fatty acid oxidation, leading to cell apoptosis and showing good therapeutic effects [9]. The high levels of CPT1 and CPT2 expressed in chronic lymphocytic leukemia (CLL) cells can lead to the death of a large number of CLL cells. Inhibiting CPT using perhexiline to inhibit the transport of fatty acids into the mitochondria has become a novel therapeutic strategy for the potential clinical treatment of CLL [10]. More and more studies have shown that the combination of targeted inhibitors of the CPT family with first-line clinical drugs shows good therapeutic prospects [11,12].

The FAO process is a metabolic pathway that facilitates the breakdown of fatty acids into CO_2_ and H_2_O and releases large amounts of energy in the body under aerobic conditions [13]. More and more studies have shown that FAO provides an important energy supply for the development of cancer cells [14]. FAs (fatty acids) are classified as short, medium, or long depending on the carbon atoms they contain, and most of the FAs found in food and body tissues are LCFAs (long-chain fatty acids) [15]. In normal physiological or pathological conditions such as fasting, prolonged exercise, and disease, where energy support is urgently needed, the oxidative supply of LCFAs is often essential, and LCFAs are indispensable to the CPT system [16,17]. LCFAs, assisted by key enzymes such as lipoyl CoA synthetase, CPT1, carnitine acylcarnitine translocase (CACT), and CPT2, are transported into the mitochondrial matrix and thus oxidized for energy supply by transforming them into fatty acids such as lipoyl CoA, fatty acyl-coenzyme A, and fatty acyl carnitine and by undergoing the steps of fatty acid activation, carnitine coupling, carrier transport, and carnitine uncoupling [18] (Figure 1).

Currently, there is growing evidence that families of CPTs are closely associated with the genesis and progression of neoplastic growth. Therefore, we look at molecular structural properties of the CPT family, the roles they play in tumorigenesis and development, the target drugs, and the possible regulatory roles of CPTs in energy metabolism reprogramming to help understand the current state of CPT family research and to search for new therapeutic strategies.

## 2. Molecular Structural Features of the CPT Subfamily

CPT1 exists within the human body in three independently encoded isoforms, CPT1A (liver subtypes), CPT1B (muscle subtypes), and CPT1C (brain subtypes), while only one form of CPT2 exists [17]. It has been confirmed that the CPT1A and CPT1B isoforms of CPT1 are situated within the external mitochondrial membrane, while CPT2 is situated within the internal mitochondrial membrane [19]. However, there is still controversy regarding the subcellular localization of CPT1C [20] (Table 1).

### 2.1. CPT1A

CPT1A, also known as CPT1-L, belongs to the hepatic isoform of the CPT1 isoform and is localized on chromosome Chr11q1p32. Although much was discovered about the CPT system in the first 15 years, CPT1 but not CPT2 is selectively inhibited by, for example, malonyl coenzyme A [21]. However, the isolation of CPT1 has been a challenge due to the fact that CPT1 binds more tightly to the outer mitochondrial membrane and is easily inactivated during elution [22]. Therefore, there has been controversy about whether CPT1 is the same as CPT2, a similar polypeptide, or a different protein [23]. In 1933, Esser et al. successfully isolated CPT1A from rat liver and identified its molecular weight as 88,150 Da [24]. Subsequently, Britton et al. localized it to the region of chromosome Chr11q 13.1 to 13.5 by FISH and other techniques [25]. The results of the cDNA assay showed that the nucleotide order of the human cDNA as well as the initial structure of the protein showed 82% and 88% similarity to those of the rat, respectively, and were essentially identical even in terms of mRNA size (~4.7 kb) [26]. Divided into three isoforms, CPT1 (CPT1A, CPT1B, CPT1C) can be divided into an N-terminal localization portion, a C-terminal substrate-catalytic portion, and an α-helix that spans the membrane twice in between [27]. CPT1A is anchored to the external membrane mainly through the hydrophobic transmembrane fragments TM1 and TM2, and studies have shown that its C-terminal end contains two malonyl coenzyme-binding sites, which are essential for a malonyl coenzyme A-sensitive conformation, while the N-terminal end has no binding site, but is able to play a regulatory role by modulating changes in the interaction of the C-terminal structural domains [28,29]. Studies of the quaternary structure of CPT1A have shown that CPT1A can function by spontaneously assembling into an oligomeric complex structure, with a trimer as the basic unit of the quaternary structure and triggered by the first 147 residues at its N-terminal end (including its two transmembrane fragments) independently of its catalytic C-terminal structural domains [30]. Furthermore, studies have shown that CPT1A is expressed in a variety of organ tissues in the human body, such as the liver, kidney, lung, ovary, pancreas, and spleen, with the highest expression in liver tissue [31,32].

### 2.2. CPT1B

Also known as CPT1-M, CPT1B belongs to the muscle isoform of CPT1 typing, contains 772 amino acids, has an average molecular weight of 88,227 Da, and is located on chromosome 22q13.3. It has been demonstrated that the nucleotide order of the human cDNA as well as the initial structure of the protein exhibit a similarity of 84.6% and 85.9%, respectively, to the rat gene [25]. The primary protein structure of CPT1B is similar to that of the liver isoform in terms of major framework features, is also anchored to the external mitochondrial membrane via its transmembrane hydrophobic segment, and contains the effective site of the C-terminal region located on the cytoplasmic side of the mitochondrial external membrane [33]. Similarly to the hepatic isoform, CPT1B functions as a rate-limiting enzyme for FAO at the mitochondrial external membrane and is inhibited by malonyl coenzyme A [34]. CPT1B and CPT1A share 62% amino acid homology; however, the two isoforms differ slightly in terms of their expression distribution in humans and in how sensitive they are to malonyl coenzyme A inhibition. In contrast to the hepatic isoform, CPT1B has a relatively low affinity for the substrate carnitine, but is more sensitive to malonyl coenzyme A (about 100 times more sensitive than CPT1A) [35]. The protein is predominantly distributed within the cardiac and skeletal muscle and is also expressed in adipocytes and white adipocytes [36]. It has been shown that the sensitivity of CPT1A to malonyl coenzyme A can be regulated by insulin and thyroid hormones, among others, and that diabetes mellitus results in higher enzymatic activity of the enzyme in question, accompanied by a reduction in its affinity for malonyl coenzyme A, whereas CPT1B is unaffected [37].

### 2.3. CPT1C

CPT1C belongs to the brain subtype of CPT1 typing and is localized on chromosome 19q13.33. Compared to the liver and muscle subtypes, CPT1C was discovered relatively late. CPT1C has high sequence similarity to CPT1A and CPT1B, all the structural motifs associated with the malonyl coenzyme A-binding site, and can bind malonyl coenzyme A at the physiological level [38]. It is noteworthy that despite the presence of all the motifs typically associated with acyltransferase activity in its primary structure, no catalytic activity has been found for lipoyl CoA, a common substrate of CPT1 [39]. In addition, the human and mouse CPT1C protein sequences share 83.5% homology, and the CPT1C sequence is slightly more homologous to CPT1A than to CPT1B (human CPT1C shares 54.5% and 52.7% homology with CPT1A and CPT1B, respectively) [40]. It is important to note that it was initially regarded as the only CPT isoform expressed within the human brain, but with the discovery of CPT1C, this claim was considered to be biased, and it was shown that CPT1C is able to act as an energy-sensing target for malonyl coenzyme A in the brain in the regulation of ingestive behavior [41]. Structural analysis of the mRNA sequence structures of the three isoforms of CPT1 showed that only CPT1C has a 5′UTR upstream open reading frame (uORF) and maintains a low level of expression by repressing the level of translation initiation of the mORF in physiological situations, while this uORF dependence can be alleviated in the presence of external stimuli, such as a decrease in glucose levels or suppression of AMPK [42]. This is a further indication of the unique role of CPT1C, as a brain subtype of CPT1, in regulating hypothalamic energy homeostasis. Although CPT1A is also localized to a small extent in the brain, unlike CPT1A, which is mainly localized to astrocytes, CPT1C brain expression is mainly restricted to neuronal cells in hypothalamic feeding centers [43]. It has also been shown that CPT1C is also expressed in human mesenchymal hepatocytes [44]. Moreover, it has been shown that the CPT1C-deficient mouse tends to have reduced food intake, lower body mass index, less body fat, and decreased fatty acid oxidation in CPT1C-knockout mice, in accordance with its function as an energy target for the detection of malonyl coenzyme A [45]. Sierra et al. further found that the function of CPT1C is neuron-specific, and its N-terminus, although devoid of mitochondrial input signals, was found to have a microsomal-aimed signaling peptide, which is mainly responsible for endoplasmic reticulum location. With the assistance of high-performance liquid chromatography–mass spectrometry, it was tentatively suggested that palmitoyl-coenzyme A is a substrate for the enzyme-catalytic properties of CPT1C [46]. Apart from its ability to regulate feeding and energy homeostasis, CPT1C has been found to have other effects. For example, overexpression of CPT1 in primary hippocampal cultured neurons has been shown to increase ceramide levels and severely impair spatial learning in its absence [47].

### 2.4. CPT2

Unlike CPT1, which has three molecular isoforms, CPT2 is located on chromosome lp32, has a molecular mass of 71 kDa, and exists only as a single isoform [48,49]. The cloning and sequencing of CPT2 was first conducted in 1991, resulting in the discovery of a 1974 bp open reading frame that encodes a 658-amino acid residue protein [50]. Unlike CPT1, which is firmly anchored to the external membrane through specific hydrophobic transmembrane portions, CPT2 is more lightly attached to the internal membrane and is easily obtained due to its low sensitivity to detergents such as malonyl coenzyme A [21]. CPT2 is structurally composed mainly of an N-terminal part and a C-terminal part, each of which consists of a six-stranded, central antiparallel b-sheet [51]. Additionally, it has been demonstrated that the nucleotide order of the human cDNA as well as the initial structure of the protein exhibit a similarity of 85% and 82%, respectively, with the rat gene [52]. As a universally distributed protein, CPT2 is more abundantly expressed in the heart, liver, skeletal muscle, etc. [23]. What is more, CPT2, like most matrix proteins, has excisable signaling molecules present at its N-terminal end that direct it to different regions for membrane-to-membrane movement by means of potential transitions across the cell membrane, whereas CPT1A’s N-terminal portion is directly present in the region to direct as well as in the program required for movement between [53]. It has been suggested that CPT1 and CPT2 may share a common site for kinetic interaction with malonyl coenzyme A-binding proteins, which enables the acquisition of sensitivity to malonyl coenzyme A [54]. Substrate specificity studies of CPT2 catalysis have shown that CPT2 was catalytically active on mid-length and long-chain acyl-coenzyme A esters; however, it was virtually inactive on short-chain and ultralong-chain acyl-coenzyme A or branched-chain amino acid oxidation intermediates [19].

**Table 1 biology-13-00892-t001:** Molecular structural features of the CPT subfamily.

Type	CPT1A	CPT1B	CPT1C	CPT2
Alternative name	CPT1-L, CPT1 liver subtype	CPT1-M, CPT1 muscle subtype	CPT1-B, CPT1 brain subtype	none
Localization of chromosomes	11q1p32	22q13.3	19q13.33	lp32
Number of amino acids	773	772	803	658
Distribution of major organizations	Liver, kidneys, lungs, brain, intestines, lungs, ovaries, pancreas, spleen, etc. [31,32].	Heart, skeletal muscle, adipocytes, and white adipocytes, among others [36].	Brain, testicles, etc. [43].	Heart, liver, skeletal muscle, etc. [23].
Functional positioning	Mitochondrial outer membrane	Mitochondrial outer membrane	Endoplasmic reticulum [41].	Inner mitochondrial membrane
Sensitivity to malonyl coenzyme A	Generally sensitive and regulated by insulin and thyroid hormones [21].	Highly sensitive, about 100 times more sensitive than CPT1A [35].	Capable of binding, but not enzymatically active in mitochondria [38].	Very low sensitivity [21].
Basic structural characteristics of proteins	A short N-terminal regulatory domain and a long C-terminal catalytic domain, two transmembrane fragments TM1 and TM2, and an intermembrane binding region connecting the two transmembrane fragments [55].			Mitochondrial lead peptides, postulated membrane interaction regions, NT and CT domains [56].
Key structural characteristics	TM1 and TM2 are responsible for anchoring the outer membrane, and the N-terminal domain can be transformed between Nα (M-CoA-sensitive) and Nβ (M-CoA-sensitive) conformations, while CPT1C is always in the Nα conformation [57].			The TM region is absent and is anchored on the inner membrane by assuming the membrane [55] interaction zone. The mitochondrial leader peptide is cleaved before entering the inner membrane [58].
Structural similarity	Nucleotide sequence and amino acid sequence similarity to rats were 82% and 88%, respectively [26].	Nucleotide sequence and amino acid sequence similarity to rats were 84.6% and 85.9%, respectively [25].	The amino acid sequence was 83.5%, similar to that of mouse [40].	Nucleotide sequence and amino acid sequence similarity to the rat were 85% and 82% respectively [52].
Sequence homology	CPT1A and CPT1B share 62% amino acid homology, CPT1A shares 54.5% amino acid homology with CPT1C, and CPT1B shares 52.7% amino acid homology with CPT1C [40].			

## 3. Role of the CPT Subfamily in Tumors

### 3.1. Role of the CPT1 Subfamily in Tumors

#### 3.1.1. CPT1A and Tumors

It is implied that CPT1A plays a pro-cancer role in most cancers, including breast, colorectal, ovarian, gastric, and lung cancers, and is closely associated with proliferation, metastasis, and drug resistance of tumor cells [59,60,61]. One of the upstream regulatory molecules of CPT1 in breast cancers with high metastasis as a therapeutic challenge is ACSL4 (long-chain acyl-coenzyme A synthase 4), which affects the FAO process and hinders therapeutic efficacy by promoting its expression [62]. In highly aggressive triple-negative breast cancers, FAO has been shown to be a key dysregulation pathway in metabolism, and inhibition of FAO is emerging as a new research hotspot [63,64]. In addition, CD24 has been shown to be an upstream regulator of the transcription factor PPARα and NF-κB signaling pathways, promoting tumor growth, proliferation, and stemness by upregulating the expression of CPT1A, thereby metabolically reprogramming the mitochondrial FAO pathway [65].

In colorectal cancer, several genes in the FAO pathway, especially CPT1A, are activated through multiple pathways and are closely associated with the metastasis of colorectal cancer cells [66]. For example, valosin-containing protein can bind to histone deacetylase 1 (HDAC1) to form the upstream transcriptional regulatory complex of CPT1A and regulate the expression of CPT1A by influencing its transcriptional modification, and the combination of targeted inhibitors of the transcriptional regulatory complex can significantly improve the therapeutic effect [67]. In contrast, Cori.ST1911, a gut microbiota isolated and identified from the intestines of mice consuming a fat-rich meal, was also able to induce acylcarnitine accumulation through upregulation of CPT1A expression, leading to activation of MEK/ERK signaling and thus promoting colorectal cancer development [68]. Recently, it has been shown that 2,6-dihydroxypeperomone B extracted from plants can selectively bind covalently to Cys96 of CPT1A and disrupt the integrity of the mitochondrial membrane by affecting its formation of the VDAC1 complex, thereby inducing apoptosis in colorectal cancer cells [59]. Increasing evidence suggests that FAO genes, represented by CPT1A, have become potential therapeutic targets in ovarian cancer, especially high-grade plasmacytoid ovarian cancer (HGSOC) [69,70]. In HGSOC, CPT1A is directly targeted by upstream TAK1 and indirectly targeted by miR-33b, and is able to promote peritoneal metastasis and dissemination of ovarian cancer cells by affecting the activation of NF-κB signaling [71].

In addition, more and more studies have shown that CPT1A can promote the growth and continuous metastasis of cancer cells to distant tissues, and even autophagy through epigenetic modification and exerting regulatory effects on downstream genes [72,73,74]. Recent studies have shown that primary breast tumors or high-fat diet-induced alveolar type 2 (AT2) cells are able to secrete palmitate, and the secreted palmitate can be used by cancer cells to acetylate nuclear factor κB (NF-κB)/p65 for providing pre-metastatic ecological niches for breast cancer lung metastasis through CPT1A and lysine acetyltransferase 2a (KAT2a) [72]. With the in-depth study of epigenetic modification, researchers have found that CPT1A can also act as a succinylation-modifying enzyme to play a role in regulating the function of cancer cells through epigenetic modification of lysine [73]. In addition, it was found that CPT1A can act in the succinylation of the lysine 302 (K302) locus of the downstream target gene MFF, inhibit its ubiquitin-proteasome degradation, and play a role in promoting cancer by affecting its expression level in ovarian cancer cells. FAO does not act as a direct influencing condition in this process [74]. Moreover, in the highly aggressive malignant hypopharyngeal squamous cell carcinoma (HSCC), it was shown for the first time that there is also a close relationship between CPT1A and autophagy, which regulates the protein expression level and affects the process of autophagy by affecting the succinylation modification of the lysine site of the ATG16L1 protein related to autophagy, thereby mediating the chemotherapy resistance of HSCC [72]. In gastric cancer, CPT1A is associated with the malignant phenotype of gastric cancer cells, and its higher protein levels tend to predict and shorten the survival time, so it can be used as a good prognostic bioindicator for the diagnosis and treatment of gastric cancer patients [75]. In addition, CPT1A can also succinylate the K222 position of LDHA and the lysine residue 47 (K47) of the calcium-binding cytoplasmic protein S100A10, inhibit the ubiquitylation proteasomal degradation of LDHA and S100A10, and consequently facilitate the malignant biological process of gastric cancer cells [76,77]. Mechanistically, it is known that the upstream regulatory pathways of CPT1A in gastric cancer include the ERK/PPAR/CPT1A and YAP/CPT1A signaling pathways, as well as the RNA regulatory network, which is directly regulated by microRNAs and circRNAs, and the circ_0024107/miR-5572/miR-6855-5p/CPT1A pathway and other signaling axes composed of microRNAs and circRNAs, by regulating FAO metabolic reprogramming, which in turn promotes lymphatic metastasis of gastric cancer cells [78,79,80,81,82].

In addition, there is growing evidence that CPT1A expression is closely linked to tumor immunity [83,84,85]. Studies have shown that PD-1 is able to promote endogenous fatty acid oxidation by increasing CPT1A expression, thereby altering the metabolic reprogramming of T cells [83]. In turn, T cell-derived IFN-γ induces upregulation of FAO and CPT1A expression in an AMPK-dependent manner, thereby conferring cancer cells immunologically mediated cytolytic killing capacity [85]. The same study found that CPT1A was able to induce MAVS Cys79-palmitoylation stabilization and activation, whereas chemotherapy targeting CPT1A could significantly enhance the application of epigenetic therapy in the field of cancer [86].

In acute myeloid leukemia (AML), CPT1A inhibitors have shown significant anti-leukemia cell activity, and patients with lower expression levels have shown longer survival times [87]. Interestingly, however, in contrast to the previous studies, CPT1A expression was low in chronic myeloid leukemia, with expression lower than 50-fold of normal, and its expression increased after treatment with tyrosine kinase inhibitors, but the detailed reasons and mechanisms of low expression of CPT1A in chronic myeloid leukemia are still unclear [88,89].

#### 3.1.2. CPT1B and Tumors

A large number of studies suggest that CPT1B takes part in promoting the malignant progression of cancer cells in a variety of cancers, such as breast cancer, gastric cancer, prostate cancer, pancreatic ductal adenocarcinoma, etc., and is closely related to the malignant phenotype of tumor cells and ferroptosis [90,91,92,93,94,95]. In high-grade invasive bladder cancer and acute myeloid leukemia, the protein level of CPT1B in cancer tissues is lower than that of adjacent cancer and can promote the malignant biological behavior of cancer cells and prolong the survival of patients [96,97]. For example, in breast cancer cells represented by breast cancer stem cells and chemotherapy-resistant cells, there is a mammary adipocyte-derived leptin-LEPR-JAK-STAT3 signaling axis, and CPT2 is directly upregulated by the upstream transcription factor STAT3, which regulates chemotherapy sensitivity through the activation of the FAO pathway [91]. Additionally, in gastric cancer, adipocytes induce high expression of phosphatidylinositol transfer protein PITPNC1, and the highly expressed PITPNC1 was able to promote drug resistance and gastric cancer omental metastasis by facilitating the nuclear translocation of PPARγ in order to increase the expression of CPT1B, which enhances FA uptake and oxidation [90]. Another study found that highly expressed guanylate cyclase-binding membrane receptor NPRA bound to PPARα and prevented PPARα degradation, and under the protection of NPRA, the expression of PPARα was increased to activate CPT1B, which promoted fatty acid oxidation and malignant biological behavior of gastric cancer cells [98].

Similarly, research suggests that the expression level of CPT1B is higher in cells resistant to oxaliplatin compared to non-resistant strains of gastric cancer, whereas the combination of the CPT1B inhibitors pegfilgrastim and oxaliplatin was able to promote apoptosis of gastrointestinal cancer cells and delay the progression of gastrointestinal cancers [94]. In addition, in chemotherapy-resistant breast cancer cells, ERRγ, a subtype of estrogen receptor-associated receptors (ERRs), is upregulated due to the mRNA splicing of the precursor ESRRG triggered by N6-methyladenosine (m6A), and interactions between the high expression of ERRγ and p65 enhanced the transcription of CPT1B, which in turn facilitated FAO to mediate chemotherapy cancer cell resistance [93]. Moreover, in castration-resistant prostate cancer, CPT1B plays a role in promoting cancer, and the androgen receptor (AR) has a direct binding target in its promoter region, which can directly inhibit its expression by transcription, while the increased expression level of CPT1B can promote the castration resistance of prostate cancer cells, as well as proliferation and migration, by activating AKT phosphorylation [92]. Studies have shown that in lung adenocarcinoma, CPT1B is subjected to direct transcriptional repression by the transcription factor MITF, and its high expression is able to facilitate stemness in lung adenocarcinoma cells by activating FAO [95]. Currently, the latest investigation has shown that CPT1B is also closely linked to the occurrence of ferroptosis. For example, in pancreatic ductal adenocarcinoma, CPT1B affects the expression and silencing of genes related to ferroptosis downstream of KEAP1 through protein interaction with KEAP1, thereby inhibiting the occurrence and development of ferroptosis, and the combined use of gemcitabine and knockdown of CPT1B can significantly enhance the therapeutic effect of the chemotherapy drug gemcitabine [99].

The majority of surveys have demonstrated that CPT1B exerts a pro-oncogenic function in a multitude of cancers; however, CPT1B also exhibits an oncogenic role in some cancer types. For example, low expression of CPT1B in high-grade muscular invasive bladder cancer (MIBC), and ectopic high expression of CPT1B can increase FAO and impede the proliferation and colonization and metastasis of bladder cancer cells [96]. Similar to cancer cells, leukemia cells, including chronic myelogenous leukemia (CML), prefer to use the glucose metabolic cycle for energy rather than oxidative metabolism. A study showed that the expression rates of CPT1A and CPT1B were 50 times lower than normal in long-term bone marrow cultures (TBMC) of CML patients [88]. In contrast, a study of 324 patients with acute myeloid leukemia (AML) found that CPT1B was significantly higher in AML patients than in the general population, and 324 clinical patient cases showed that this result often indicated shorter survival [97].

#### 3.1.3. CPT1C and Tumors

Unlike other members of the CPT family, CPT1C is poorly understood, being the most recently discovered brain-specific subtype among the members [38]. A substantial body of evidence from numerous studies indicates that CPT1C signaling plays an important role in feeding behavior and systemic energy utilization within the hypothalamus [100]. It was not until 2011 that Zaugg et al. first revealed that CPT1C could inhibit the mTOR pathway and tumor reactivity to rapamycin in human breast cancer and lung cancer, and under external regulatory stimuli, such as reducing glucose concentration or a low-oxygen environment, CPT1C protein expression level could be promoted in an AMPKα-dependent manner and apoptosis could be inhibited, whereas knockdown of CPT1C was able to delay breast and colon cancer-derived tumor growth and metformin responsiveness in vivo [101]. Subsequently, more and more studies have demonstrated that CPT1C plays an important pro-carcinogenic role in a variety of cancers, such as colorectal, endometrial, hepatocellular, gastric, esophageal squamous cell, and breast cancers [102,103,104,105].

In colorectal cancer, it has been found that the upstream transcription factor of CPT1C is HIF1α, which transcriptionally promotes CPT1C by binding directly to the promoter region of CPT1C, thus exerting functions such as promoting the proliferation, migration, accelerating cell cycle progression, and FAO rate of colorectal cancer cells [104]. A direct transcriptional upregulation of CPT1C by the hypoxia-inducible factor HIF1α was similarly found in gastric cancer and was associated with the malignant proliferative behavior of gastric cancer cells [106]. In contrast, in gastric cancers that developed ovarian metastases, CPT1C was similarly found to be able to promote the invasive growth of gastric cancer cells toward the ovarian site by upregulating the FAO rate and was able to increase the stemness characteristics of the cells [102]. Furthermore, in endometrial cancer, CPT1C is regulated by upstream acyl-coenzyme A synthetase long-chain family member 1 (ACSL1) and is able to promote malignant phenotypes such as proliferation, migration, and so on by affecting the FAO rate [105]. In thyroid-like carcinoma, AMPK activates and protects against hypoxia- and hypoglycemia-induced cancer cell death by influencing the level of CPT1C expression in response to stimuli such as external anthropogenic interventions lowering oxygen concentration or glucose levels [107]. Nonetheless, in hepatocellular carcinoma, a direct target binding site for MiR-377-3p exists in the 3′UTR region of CPT1C, which affects the downstream FAO rate through direct target inhibition, thereby influencing the malignant biological behavior of hepatocellular carcinoma [103]. Similarly, in esophageal squamous cell carcinoma, CPT1C was able to directly affect FAO rate and cell cycle G1/S transition, exerting a pro-carcinogenic effect by maintaining intracellular redox homeostasis as well as energy supply levels [20]. Recently, it was found that CPT1C can act as a novel target gene of ERRα and regulate the proliferation, metabolism, and tumorigenesis of breast and pancreatic cancer cells through the direct regulatory network formed by miR-1291, ERRα, and CPT1C [108]. In addition, CPT1C is also subject to direct transcriptional repression by the transcription factor ZEB2 and regulates FAO activity through a regulatory network formed by mutant p53 (Mutp53) and miR-200c, which promotes migration, invasive phenotypes, and cell stemness in basal-like breast cancer cells [109]. The same study highlighted that CPT1C downregulation led to breast cancer stromal remodeling and anthracycline resistance, suggesting that CPT1C could serve as a novel predictive biomarker for breast cancer chemotherapy [103].

A growing number of investigations have emphasized that CPT1C likewise plays an important role in sustaining the senescent phenotype of cells, and the regulatory relationship between CPT1C and cellular senescence predicts the ability to become a new drug target for epigenetic therapy [110,111]. In neurofibromas, it was found that p53 is also an upstream regulator of CPT1C that can modulate the metabolic network of cancer cells by influencing changes in its expression level, and in vivo experiments have shown that in CPT1C-knockout mouse tumor models, the rate of tumor growth is relatively retarded, as well as the number of days of survival being significantly increased [112]. Both CPT1A and CPT1B isoforms of CPT1 are now known to function as downstream target genes for PPARα regulation in many cancers [113]. Recently, findings in breast cancer have implied that CPT1C, also a PPARα target gene, functions to regulate the senescence phenotype of breast cancer cells and that this regulatory relationship is not directly related to the presence or absence of P53 [114]. Moreover, the knockdown of CPT1C in human pancreatic epithelioid carcinoma also affects mitochondrial function and disrupts cellular energetic homeostasis, thereby contributing to the senescent phenotype of the cancer cells [110]. Further, a study comparing other CPT isoforms (CPT1A, CPT1B, and CPT2) concluded that except for CPT1C, other members of the CPT family have not been found to be associated with cancer cell senescence for the time being, and suggested a potential molecular mechanism by which CPT1C/c-Myc/p27 contributes to the induction of cellular senescence by CPT1C knockdown [115]. Reduced CPT1C expression levels were found to significantly increase the progression of the senescence phenotype in breast cancer cells by 13C-metabolic flux (13C-MFA) analysis, and stearate inhibited cell proliferation, whereas oleate reversed the silencing of the CPT1C-induced senescence phenotype [116]. Further studies have shown that CPT1C is not only associated with cellular senescence but also closely linked to the regulation of cancer cell lipotoxicity and tumor immunity, which contributes to a deeper understanding of the importance of CPT1C as a therapeutic target [117]. A new immunosuppressive subpopulation of fibroblasts exists in gastric cancer, which can exclusively overexpress CPT1C and secrete the pro-inflammatory cytokine IL-6 to interfere with the immunosuppressive function of macrophages in gastric cancer, thereby exerting a pro-cancer effect [118].

Currently, the majority of studies have demonstrated that despite CPT1C exhibiting suboptimal catalytic activity and being situated within the endoplasmic reticulum, it still affects the malignant biological learning of cancer cells by altering downstream FAO rates [103]. However, by integrating the available data, some researchers have found that the relationship between CPT1C and FAO rate may not be a direct influence, but rather plays an indirect regulatory role by acting as an inductive mediator affecting the expression or function of SAC1, protruding proteins, ABHD6, ABHD12, etc., and thus functioning to promote malignant progression of tumor cells [119]. In addition, the multisubunit E3 ubiquitin ligase APC/C was found to regulate the protein level of CPT1C, which was associated with the presence of its binding and recognition sequences and the KEN box and the D-box motifs, which have not been found in other isoforms [20]. This suggests that our study of the molecular regulatory mechanisms of CPT1C could lead to a more in-depth study of its protein level regulation. Current evidence underscores that CPT1C plays a predominantly pro-cancer role in most cancers [120]. Curiously, however, for cells where sufficient energy and material commonality already exist, for example, it is interesting to note that the expression of CPT1C in cells with unlimited nutrients is instead relatively low [121]. The exact mechanism of this phenomenon is still unclear. The CPT1C protein is susceptible to inhibition by small-molecule compounds, which are capable of acting as enzymes. Additionally, due to the existence of the blood–brain barrier, it is difficult to maximize the efficacy of most natural compounds or small-molecule drugs for specific molecular targeting, and this characteristic of CPT1C makes it a great therapeutic prospect for specific targeting [100].

### 3.2. Dual Role of CPT2 in Tumors

#### 3.2.1. Carcinogenesis

Recent studies have emphasized that CPT2 is pro-carcinogenic in chronic lymphocytic leukemia (CLL), epithelial ovarian cancer (EOC), gastrointestinal cancer (GIC), TNBC, etc., and that it plays an important role in promoting proliferation, migration, invasion, chemo-resistance, and radiotherapy resistance of cancer cells [10,11,94,122,123]. For example, in chronic lymphocytic leukemia (CLL), CPT2 is highly expressed, and the inhibitor of CPT2 pegfilgrastim is able to inhibit FA transport into mitochondria at clinically achievable concentrations, leading to the death of numerous chronic lymphocytic leukemia cells, while lymphocytes and stromal cells are less affected [10]. Nonetheless, in epithelial ovarian cancer, the developmental regulator NKX2-8 is able to transcriptionally inhibit CPT2 by recruiting the transcriptional repressor complex Sin3A/HDAC1/SAP18, and the combination of pegfilgrastim and platinum drugs enhances therapeutic effects in epithelial ovarian cancer [11]. Similarly, studies have shown that knockdown of CPT2 or the use of FAO inhibitors in recurrent breast cancer significantly inhibited ERK expression and cell stemness in radiotherapy-resistant cells, thereby improving the efficacy of radiotherapy in recurrent breast cancer cells [122]. In contrast, in patients with gastrointestinal cancers that were poorly treated with oxaliplatin, the combination of perhexiline, an inhibitor of CPT2, and oxaliplatin was able to significantly reduce cancer cell survivability from chemotherapy via the ROS/NFATc3/CPT2 axis [94]. In contrast, in TNBC resistant to treatment with the glutaminase inhibitor CB-839, inhibition of CPT2 significantly enhanced its therapeutic efficacy and reduced the proliferation and migration of TNBC cells [123]. In addition, HMG-CoA reductase degradation protein 1 (HRD1) was found to be the E3 ligase of CPT2, leading to high CPT2 expression in TNBC by promoting degradation of its protein ubiquitination level [124]. Numerous findings have emphasized that CPT2 is closely associated with malignant biological processes in a wide range of cancers (Table 2). The use of CPT2 inhibitors in conjunction with clinical first-line drugs has a favorable combination effect, and CPT2 may be a good drug target for molecularly targeted therapies.

#### 3.2.2. Cancer Inhibition

Numerous studies have shown that CPT2 is oncogenic in hepatocellular carcinoma, colorectal carcinoma, renal clear cell carcinoma, and primary ovarian plasmacytoid carcinoma, and is associated with proliferation, migration, invasion, stemness, apoptosis, and chemo-resistance of cancer cells [125,126,127]. Interestingly, research has emphasized that CPT2 is significantly overexpressed and pro-carcinogenic in epithelial ovarian cancer, whereas it is underexpressed and shows reduced long-term survival in another type of ovarian cancer, primary ovarian plasmacytoid carcinoma, whereas low levels of CPT2 promote apoptosis through the NADPH/ROS/NFκB signaling pathway [11,126]. In contrast, in colorectal cancer, a regulatory network formed by SGMS1-AS1 and microRNA-106a-5p exists upstream of CPT2, leading to a decrease in CPT2 expression and enhancement of glucose uptake and lactic acid production, resulting in an immunosuppressive tumor microenvironment that promotes the development of colorectal cancer [128]. Another study on colorectal cancer showed that downregulated CPT2 is equally capable of promoting colorectal cancer stemness and oxaliplatin resistance by disrupting the balance of intracellular redox homeostasis, promoting ROS production and activating the downstream Wnt/β-catenin signaling pathway, and inducing glycolytic metabolism [129]. In addition to inducing glycolytic metabolism, it has been shown that p53 can also act as a downstream target gene of CPT2 and play a role in promoting proliferation, migration, and invasion, as well as apoptosis, by affecting its activity in colorectal cancer cells [130]. For renal clear cell carcinoma, CPT2 was able to promote the malignant biological behavior of cells, as well as increase sorafenib resistance by inhibiting FAO and decreasing NADPH (nicotinamide adenine dinucleotide phosphate), thereby activating the ROS/PPARγ/NF-κB pathway [127].

Compared with other colorectal, ovarian, and renal clear cell carcinomas, the mechanism of CPT2 in hepatocellular carcinoma has been relatively well studied, and the studies have placed more emphasis on upstream transcription factors, inhibitors of CPT2, and the effects of combinations of drugs. Scientific findings emphasize that CPT2 is pro-cancer in hepatocellular carcinoma and is significantly correlated with tumor histological differentiation and venous infiltration. Knockdown of CPT2 can promote adipogenesis in cancer cells through upregulation of stearoyl coenzyme A desaturase 1 (SCD1), which significantly enhances hepatocellular carcinoma tumorigenicity, metastatic potential, and cisplatin resistance [125]. In contrast, PPARα exhibited a low level of expression, and transcriptional promotion of CPT2 in hepatocellular carcinoma driven by obesity and nonalcoholic steatohepatitis could partially explain the downregulation of CPT2 expression, which was shown to be able to lead to hepatocellular carcinoma cell resistance to lipotoxicity through inhibition of Src-mediated JNK activation and to play a pro-cancer role through acylcarnitine accumulation exerting pro-cancer effects [131]. The same study found that E2F1 and E2F2 could also act as upstream transcriptional repressors of CPT2 in nonalcoholic steatohepatitis-induced hepatocellular carcinoma and that low expression of CPT2 could provide the lipid-rich environment required for hepatocarcinogenesis [132]. Orlistat, a gastrointestinal FA synthase inhibitor, is efficacious in promoting apoptosis and inhibiting the proliferation of many types of cancer cells [133]. Studies have shown that orlistat treatment reduces the enzyme activity and expression of CPT2 and has a stronger therapeutic effect on growth inhibition and apoptosis in hepatocellular carcinoma after combination treatment with paclitaxel [134]. Metabolomic analysis of hepatocellular carcinoma cells showed that CPT2 was significantly downregulated in HCC samples, and the downregulation of CPT2 resulted in a notable increase in the levels of LCFAs, a significant decrease in the levels of short- and medium-chain FAs, and a clear inhibition of the carnitine shuttle system [135] (Figure 2).

**Table 2 biology-13-00892-t002:** Role of CPT subfamily in tumors.

CPT Members	Functionality	Type of Cancer
CPT1A	cancer-promoting	Breast cancer [62,65,73]; Colorectal cancer [68,136]; Ovarian cancer [71,74]; Squamous cell carcinoma of the hypopharynx [72]; Gastric cancer [78,79,81]; lung cancer [84].
	cancer prevention	Leukemia [87,88,89].
CPT1B	cancer-promoting	Breast cancer [91,93]; Gastric cancer [90,94]; Prostate cancer [92]; Adenocarcinoma of the lungs [95,99].
	cancer prevention	Bladder cancer [96]; leukemia [88,97].
CPT1C	cancer-promoting	Colorectal cancer [104]; Gastric cancer [102,106,118]; Endometrial carcinoma [105]; Thyroid carcinoma [107]; Hepatocellular carcinoma [103]; Esophageal squamous cell carcinoma [20]; pancreatic [110,116]; Breast cancer [103,108,109,114].
CPT2	cancer-promoting	Leukemia [10]; Ovarian cancer [11]; Breast cancer [122,123]; Gastric cancer [94].
	cancer prevention	Ovarian cancer r [126]; Colorectal cancer [128,129,130]; Clear cell renal carcinoma [127]; Hepatocellular carcinoma [131,132,134].

## 4. Progress of Research on CPTs in Targeted Therapy

### 4.1. Targeted Inhibitors of CPTs

#### 4.1.1. Etomoxir

Etomoxir, alias (R)-(+)-etomoxir, is a cell-permeable, stereospecific compound [137]. As an irreversible inhibitor of CPT1A and CPT1B, it was originally developed as an oral hypoglycemic agent [138]. After ingestion, etomoxir is first hydrolyzed in vivo to free acid, which is next converted to coenzyme A ester, which blocks FAO by irreversibly competing for binding to the catalytic site on CPT1 and thereby exerting an inhibitory effect on CPT1 function [139]. Etomoxir has been shown to similarly reduce ketogenesis and protect the myocardium from ischemic damage caused by FAs [140]. In addition, studies have shown good antiproliferative effects and good activity as a hypoglycemic compound [141]. However, since etomoxir does not discriminate between CPT1A and CPT1B, it was found that long-term inhibition of CPT1B resulted in increased intracellular lipids and insulin resistance in rats [142]. It has been used in phase II clinical trials with hepatotoxicity, producing side effects such as cardiac hypertrophy, and is therefore not considered a clinical agent [139].

#### 4.1.2. ST1326 ([R]-N-[Tetradecylcarbamoyl]-aminocarnitine)

ST1326, alias teglicar, is a long-chain carbamoyl aminocarnitine derivative that selectively inhibits CPT1A while only slightly affecting CPT1B, and this inhibition is reversible [143]. Moreover, the crystal structure of rCPT-2 isolated from rats and the substrate analogue ST1326 complex showed that ST1326 could also inhibit the activity of CPT2 [144]. Due to only transient toxic effects on the liver and its anti-ketotic and antidiabetic activity, ST1326 is currently in clinical studies [145]. The ST1326 compound has been demonstrated to diminish the processes of ketone body production and glucose appearance in freshly extracted hepatocytes derived from rats that have been subjected to fasting. Furthermore, it has been evidenced to curtail gluconeogenesis and enhance glucose homeostasis in animal studies [146]. In Burkitt’s lymphoma, studies have emphasized the strong efficacy of ST1326 in hindering c-myc-induced lymphomagenesis and have found that ST1326 blocks not only CPT1 but also CACT activity, and thus further differentiation of the active blocking targets is needed to avoid unforeseen side effects when investigating the targeting effect of ST1326 [145]. In leukemia, especially acute myeloid leukemia, ST1326 can effectively inhibit the effect of FAO on leukemia cells, which leads to cell growth arrest, mitochondrial damage, and apoptosis [147]. Moreover, ST1326 has strong synergistic inhibition with chemotherapeutic drugs such as the Bcl-2 inhibitor ABT199 on AML, which may result in an enhancement of its anti-proliferative and pro-apoptotic effects [148,149].

#### 4.1.3. Perhexiline

Perhexiline, also known as perhexilene, is a calcium antagonist that inhibits Ca^2+^ inward flow, dilates vascular smooth muscle, and increases coronary blood flow, and was therefore initially invented as an antianginal drug [150]. Due to its ability to prevent diastolic dysfunction during myocardial ischemia, it is most commonly used in patients with refractory angina pectoris in whom other clinical first-line drugs are ineffective or unavailable due to certain contraindications [151]. However, because of adverse effects such as hepatotoxicity and neurotoxicity, it is currently only used as a prescription drug in some countries [150]. Fortunately, perhexiline was also found to target CPT1, improving myocardial energetics in hypertrophic cardiomyopathy (HCM) and potentially reducing left ventricular hypertrophy (LVH) in HCM [152]. Furthermore, in vitro experiments showed that perhexiline inhibited CPT1 produced by rat heart and liver mitochondria in a concentration-dependent manner, with ic50s of 77 and 148 μmol/L, respectively, and that the inhibitory effect of perhexiline on cardiac CPT1 was higher than that on liver CPT1 [153]. Moreover, in CLL (chronic lymphocytic leukemia), perhexiline was found to selectively kill CLL cells expressing high levels of CPT1 and CPT2, with little effect on other cells, and a CLL transgenic mouse model further demonstrated in vivo that perhexiline injections significantly prolonged the survival of the animals [10].

#### 4.1.4. Amiodarone

Amiodarone is a benzofuran derivative that was also developed as a prophylactic angina pectoris drug [153]. Several studies have shown that this drug has minimal effect on negative inotropy, reduces the incidence of arrhythmia, has potentially prolonged action, and blocks multiple ion channels [154]. It is currently the first-line clinical agent for the treatment of ventricular and atrial arrhythmias [155]. However, studies have found that long-term use of amiodarone can impair normal thyroid function and lead to liver abnormalities and decreased lung activity, among other serious side effects [156]. In addition to being an antianginal agent, amiodarone has been found to inhibit cardiac CPT1; however, this inhibitory utility is not strong [157]. Investigation of patients with atrial fibrillation treated with amiodarone revealed that low doses of amiodarone significantly reduced the size of glioblastoma multiforme xenograft tumors and showed strong antiangiogenic effects [158]. In contrast, desethylamiodarone (DEA), the main metabolite of amiodarone, was found to have direct mitochondrial effects involved in its cytostatic effects in melanoma cells [159].

#### 4.1.5. Other Drugs

Other antianginal drugs such as trimetazidine and ranolazine have also been found to inhibit CPT1. However, trimetazidine is not as effective as perhexiline or amiodarone in inhibiting CPT1 and is relatively less potent [160]. In contrast, S-15176, a derivative of trimetazidine, was found to noncompetitively inhibit CPT1 and had a stronger inhibitory effect on the heart than on the liver [161]. Ranolazine, a piperazine derivative, is able to increase myocardial perfusion by reducing diastolic wall tension and was initially used to treat angina pectoris and myocardial ischemia before it was found to have an effect on the FAO process in the heart [162]. It is currently used in the treatment of patients with chronic angina pectoris due to its ability to act as an inhibitor of late cardiac sodium currents and to block currents including the hERG/Ikr K ionic current [163]. Studies have shown that ranolazine is effective in the long-term treatment of patients, and as monotherapy or in combination with other drugs, it has shown relatively good antianginal effects with little or no effect on clinical hemodynamic effects [164] (Table 3).

## 5. Potential Importance of CPTs in FAO, Reprogramming of Tumor Energy Metabolism

### 5.1. CPTs Can Promote FAO and Activate Oxidative Phosphorylation

Oxidative phosphorylation is a coupling reaction in which the energy produced when a substance is oxidized in the mitochondria is supplied through the respiratory chain to ADP and inorganic phosphate to synthesize ATP [165]. Compared with normal cells, cancer cells are metabolically heterogeneous, showing not only abnormal glucose metabolism but also large changes in lipid metabolism, such as FA metabolism [166]. Reprogramming of lipid metabolism is one of the fundamental features of cancer cells and an integral part of the field of cancer metabolism [167]. FAO was found to provide not only ATP to promote the malignant biological behavior of cells but also NADPH to regulate redox state and promote tumor cell growth and malignancy [168]. Although studies have shown that energy supply via oxidative phosphorylation is maybe the most advantageous option for tumor cells because of the presence of metabolic reprogramming in tumor cells, the existence of this situation does not mean that oxidative phosphorylation is no longer important [169]. Numerous studies have emphasized that tumor cells still retain relatively intact mitochondria and the ability to oxidatively phosphorylate and that under specific conditions, tumor cells can be stimulated to use mitochondrial oxidative phosphorylation for energy supply [170]. Studies have shown that FAO-related enzymes are increased and upregulated in many tumors [171]. FAO is dysregulated in a wide range of human malignancies, and the proliferation, stemness, and drug resistance of cancer cells are dependent on FAO being normal [172]. Increased fatty acid oxidation has been found to provide mitochondria with more reducing equivalents, leading to activation of OXPHOS and increased ATP synthesis [169].

### 5.2. CPTs Are Important Regulators of Energy Metabolic Reprogramming

In addition, in the early stage of metabolic reprogramming, there is a metabolic switch where CPT1A remains highly expressed until 7 days, while CPT1A expression is downregulated and CPT1B remains at a higher level afterward. Subsequently, high expression levels of CPT1B promoted FAO, which was able to facilitate reprogramming through upregulation of oxidative phosphorylation (OXPHOS) and downregulation of protein kinase C activity [173]. Studies have shown that mitochondrial FAO is functional in cancer cells and can drive OXPHOS-dependent ATP supply to promote tumor proliferation [174]. This suggests a deeper cross talk between FAO processes and OXPHOS processes and that this cross talk is importantly linked to CPT family members. Furthermore, research has demonstrated that carnitine, which takes part in the oxidation of LCFAs, can act as an intermediate between peroxisomes and mitochondria, forming cross talk between the two, and that carnitine enables metabolite exchanges between peroxisomes and mitochondria, which are inextricably linked to CPT family members [175]. Recent reports have shown that CPT2, a member of the CPT family, is lactylated by the mitochondrial alanyl-tRNA synthetase (AARS2) lysine 457/8 in protein lysyl lactate transferase under hypoxic conditions and that its lactylation is able to inhibit OXPHOS by restricting fatty acid oxidation, whereas the CPT2 lactylation, when reversed by SIRT3, is able to reactivate OXPHOS [176]. This suggests that the protein modification status of CPT family members represented by CPT2 can influence the cross talk between fatty acid oxidation and oxidative phosphorylation, whereas the existence of the same role for members other than CPT2 has not yet been reported and further studies are still needed(Figure 3).

## 6. Conclusions

Increasingly numerous studies have demonstrated the efficacy of intensive study of lipid metabolic reprogramming, especially long-chain fatty acid oxidation, which opens up potential discoveries for targeted therapies in malignant tumors. Studies have demonstrated the efficacy of intensive study of lipid metabolic reprogramming, especially long-chain fatty acid oxidation, which opens up potential discoveries for targeted therapies in malignant tumors. The key rate-limiting enzymes of the FAO pathway, represented by the CPT family, play important roles in tumor proliferation, migration, invasion, stemness, apoptosis, chemoresistance, aging, and metabolic stress. At present, the regulatory mechanisms and signaling pathways involved in the CPT family are the main focus of upstream studies, as the classical signaling pathways Wnt/β-catenin, NF-κB, mTOR, or transcription factors, miRNAs, etc., affect the expression of CPT family members at the transcriptional level and then play a role in promoting or suppressing cancer by affecting the FAO process, while there are relatively few studies on their protein levels. In addition, recent studies have found that CPT1A has succinyltransferase activity, which can affect the succinylation modification of downstream target molecules and affect its ubiquitin proteasome degradation to play a downstream regulatory role, so whether other members of CPTs also have the same enzyme activity needs to be further studied. There have been studies reporting that the expression of CPT1C is correlated with the senescence phenotype of cancer cells, which also provides a new research angle for the phenotypic study of other CPTs. Moreover, the current research on the CPT family focuses more on CPT1A and CPT1B, while relatively few studies have been conducted on CPT1C and CPT2 in tumors. Unlike other members of the CPT family, CPT1C is not thought to have the capacity to catalyze the transfer of acyl groups from fatty acyl CoA to carnitine, but its primary structure contains all the motifs required for known acyltransferase activity, suggesting that CPT1C may have other unique catalytic substrates that are different from other CPTs. The possible substrate reported so far is palmitoyl-CoA, but further experimental verification is still needed, and further research on CPT1C catalytic substrates in the future may help to elucidate the specific relationship between CPT1C and carcinogenesis and progression. Research on CPT2 in recent years has mainly focused on the phenotypes of CPT2, mainly on its transcriptional regulation, proliferation, apoptosis, metastasis and drug resistance, but there have been few studies on its protein level, especially protein stability. At present, it has only been reported that HRD1 is the E3 ligase of CPT2 in triple-negative breast cancer, and whether there are other E3 ligases or deubiquitinating enzymes in CPT2 is still unclear. It has been reported that CPT2 plays the opposite role of promoting or suppressing cancer in different tissue types (epithelial, serous) in ovarian cancer, suggesting that CPT2 does only plays a single pro-cancer or tumor suppressor role in different cancer types, but has some more refined switching mechanism. The regulation of its transformation in the role of cancer promotion and tumor suppression requires further research. In addition, the specific reasons and mechanisms for the low expression of CPT1A in chronic myeloid leukemia are still unclear. Most of the studies on CPT family members in tumors have focused on the transcriptional level and the phenotypic level, such as proliferation, invasion, stemness, chemoresistance, etc. It is still unclear whether the protein level and the aging phenotype represented by CPT1C are also applicable to other members of the CPT family in other types of cancers. Moreover, it has been shown that the protein modification status of CPT2, a member of the CPT family, affects the cross talk between fatty acid oxidation and oxidative phosphorylation and that CPTs can promote FAO and activate oxidative phosphorylation. The potential of CPTs as important regulators of energy metabolism reprogramming needs to be explored. Although there are already inhibitors targeting the CPT family, most of them do not strictly differentiate between CPT family members, and despite certain structural similarities, are prone to unpredictable side effects due to their wide distribution in vitro, and few of them make it to the clinical trial stage. In the discussion section, we added the need for the development of drugs that target CPT and synergy with cancer treatment. This mainly refers to the current CPT inhibitors being mostly developed as antianginals, with large side effects, and lack of clear differentiation of the classification of CPT family members. The combination of specific targeted inhibitors of CPT family members with current clinical drugs can increase drug sensitivity and reduce drug resistance, and has good application prospects. In contrast, CPT1C protein as an enzyme can be relatively easily inhibited by small-molecule compounds and due to brain-specific expression. The majority of small-molecule drugs exhibit limited capacity to traverse the blood–brain barrier, in addition to other distinctive characteristics, making them a great therapeutic prospect for specific small-molecule inhibition. In conclusion, the potential importance of CPT family members in oncology needs to be further investigated.

## Figures and Tables

**Figure 1 biology-13-00892-f001:**
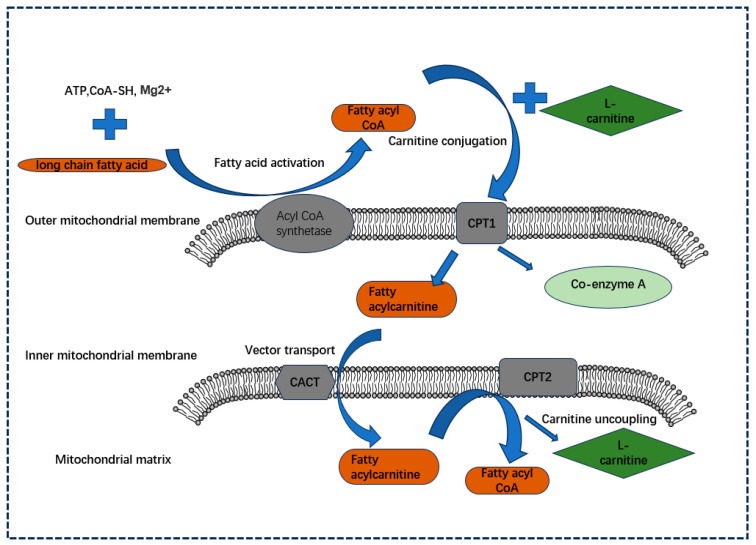
The role played by CPT family members in the oxidation of LCFAs. In the presence of Mg^2+^, CoA-SH, and ATP, LCFAs undergo fatty acid activation, a process catalyzed by lipoacyl CoA synthetase localized at the outer membrane of the mitochondria, which generates lipoacyl CoA. Subsequently, with the help of CPT1 localized at the outer mitochondrial membrane, they are changed into fatty acylcarnitine in conjunction with L-carnitine, thus entering the mitochondrial outer membrane (carnitine coupling). Subsequently, the molecule is transported from the outer to the inner mitochondrial membrane by CACT, which facilitates carrier transport, and next uncoupled back to fatty acyl-coenzyme A and L-carnitine by CPT2 localized at the surface of the inner mitochondrial membrane (carnitine uncoupling) to enter the mitochondrial matrix for the next step of β-oxidation for the production of energy and other substances.

**Figure 2 biology-13-00892-f002:**
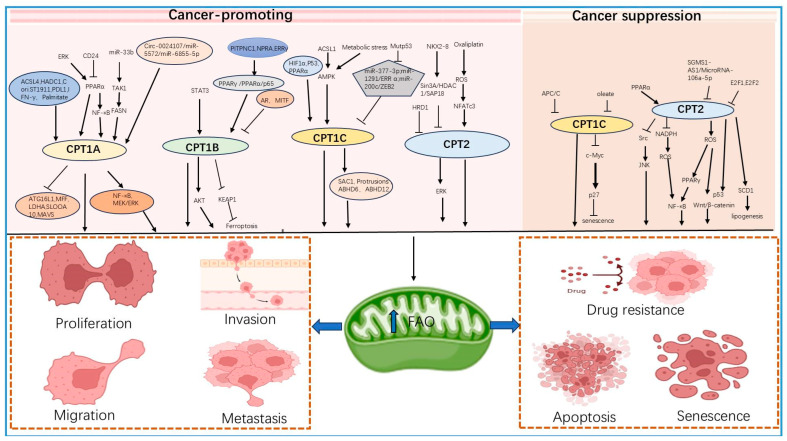
Mechanism diagram of CPT family promoting tumorigenesis and development. CPT1A, CPT1B, CPT1C, and CPT2 of the CPT family affect FAO to exert a pro-cancer effect through a variety of pathways, affecting the proliferation, migration, invasion, metastasis, drug resistance, and senescence of cancer cells, while the tumor suppression mechanism of the CPT family is currently only reported for CPT1C and CPT2.

**Figure 3 biology-13-00892-f003:**
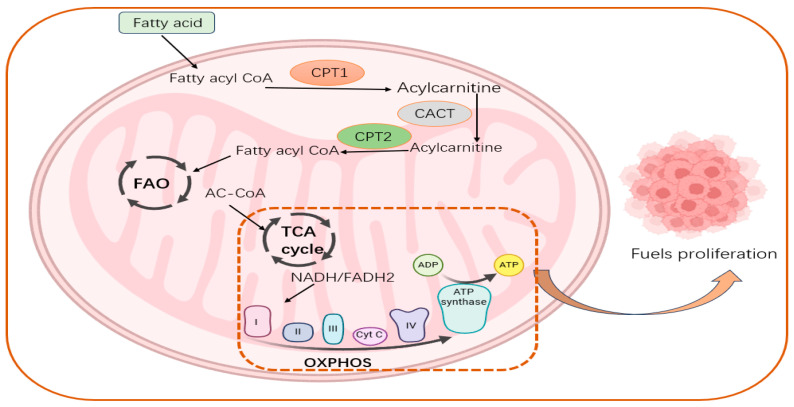
The role of the CPT family in fatty acid oxidation, oxidative phosphorylation, and tumor progression. Fatty acyl CoA entering the mitochondria is converted to acylcarnitine by CPT1, which is located in the outer mitochondrial membrane, followed by acylcarnitine by CACT, and finally to fatty acyl CoA by CPT2 for fatty acid oxidation. The product AC-CoA produced during the oxidation of fatty acids can enter the tricarboxylic acid cycle to promote oxidative phosphorylation, thereby producing a large amount of ATP to provide energy for tumor growth.

**Table 3 biology-13-00892-t003:** Targeted inhibitors of CPTs.

Inhibitor	Target	Side Effects
Etomoxir	CPT1A, CPT1B [138].	Hepatotoxicity, cardiac hypertrophy, etc. [139].
ST1326 (teglicar)	CPT1A, CPT1B [143]; CACT [145]; CPT2 [56].	Only transient toxic effects in the liver [145].
Perhexiline	CPT1 [152]; CPT2 [150].	Serious side effects such as hepatotoxicity and neurotoxicity [150].
Amiodarone	CPT1 [157].	Serious adverse reactions such as thyroid dysfunction, hepatic injury, and pulmonary toxicity [156].

## Data Availability

No new data were created or analyzed in this study. Data sharing is not applicable to this article.

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
