# Peer review of "The Role of the CPT Family in Cancer: Searching for New Therapeutic Strategies"

_biology, 2024, doi:10.3390/biology13110892_

Round 1

Reviewer 1 Report

Comments and Suggestions for Authors

The authors provide a detailed review of the functions and mechanisms of CPTs family members in promoting the development of different tumorigenesis. The text is well presented, sections properly articulated. Figures are clear and exhaustive. I think this article is acceptable after a minor revision. As following, I have minor comments/suggestions that I feel it would improve the quality of the review.

Point 1: Line 13: explain FAO at first appearance.

Point 2: Line 432: Are there any other molecules that are regulated by CPT1C listed in the section explaining the indirect regulatory role of CPT1C, and if so, please list.

Point 3:  Line 642:It would be better to replace the word no longer with maybe.

Author Response

Comments 1: Line 13: explain FAO at first appearance.

Response 1: Thank you for reviewing our article. We appreciate your helpful advice, and we have added corresponding explanations. Please see the revised text from line 13.

Comments 2: Line 432: Are there any other molecules that are regulated by CPT1C listed in the section explaining the indirect regulatory role of CPT1C, and if so, please list.

Response 2: Thank you for your comment. Accordingly, another possible regulated molecule is the hydrolase ABHD12. We have added this molecule to the revised text. Please see the revised text on line 432.

Comments 3: Line 642:It would be better to replace the word no longer with maybe.

Response 3: Thank you for your advice. Based on your suggestion, we've changed "no longer" to "maybe."Please see the revised text on line 642.

Reviewer 2 Report

Comments and Suggestions for Authors

The review was well organized and went into depth about the structural, biochemical, disease relevance, and therapeutic assessment of the CPT family. The review also did a good job in talking about some of the understudied CPT family members like CPT1C and CPT2. The review also went into depth about how some of the CPT family members have both cancer prevention and promoting phenotypes, giving complexity to the CPT enzymes and the importance to further study these enzymes. 

On the other hand, there are some areas that could be improved in the review. There were some grammatical and spelling errors throughout the review. For example, some include some errors in lines 96, 157,195, 214, 252, and 476. During the structural and biochemical review sections, a figure would be helpful for the reader. Specially in the structural section, some structures of the proteins would be helpful to enforce the message. Some biochemical schematics with disease relevance of each enzyme would also be helpful. Also, in Table 3 there were some targets on the table that are not described in the main text. 

Overall, this is a really helpful and important review for the field of lipid metabolism. After addressing the comments above, the review can be accepted for publication.

Author Response

Comments 1: On the other hand, there are some areas that could be improved in the review. There were some grammatical and spelling errors throughout the review. For example, some include some errors in lines 96, 157,195, 214, 252, and 476. 

Response 1: Thank you for pointing out this error. We have reworked and corrected grammatical and spelling errors in this review, especially lines 96, 157, 195, 214, 252, and 476.

Comments 2: During the structural and biochemical review sections, a figure would be helpful for the reader. Specially in the structural section, some structures of the proteins would be helpful to enforce the message.

Response 2: Thank you for your advice. Based on your suggestion, We added descriptions of the protein structures of CPT family members (e.g., the number of amino terminals of the encoded protein, basic structural features of the protein, and the key structural features). We summarized them into rows 3, 8, and 9 of Table 1.

Comments 3: Some biochemical schematics with disease relevance of each enzyme would also be helpful. 

Response 3: Thank you very much for your suggestion, based on your suggestion, we have added a biochemical schematic diagram between CPT family members and related diseases in the new manuscript (Please see Figure 2).

Comments 4: Also, in Table 3 there were some targets on the table that are not described in the main text.

Response 4: Thank you for pointing out this problem. Based on your suggestion, we have supplemented the target CPT2 of ST1326 and Perhexiline in the corresponding body of the table and cited relevant references. Please see the revised text from lines 552 to 554 and lines 583 to 584.

Reviewer 3 Report

Comments and Suggestions for Authors

Manuscript ID: biology-3267622

Type of manuscript: Review

Title: The role of the CPTs family in cancer: Searching for new therapeutic strategies

Authors: Yanxia Duan, Jiaxin Liu, Ailin Li, Chang Liu, Guang Shu *, Gang Yin*

Journal: Biology

The manuscript entitled „The role of the CPTs family in cancer: Searching for new therapeutic strategiesË® describes the CPTs family, the roles they play in tumorigenesis and potential as drug targets. The manuscript is interesting and well written. I recommend publication and suggest a minor corrections:

1.      Page 3, line 96 part: CPT1 (CPT1A, CPT1B, CPT1B) has two times CPT1B change to CPT1 (CPT1A, CPT1B, CPT1C)

2.      Check through the manuscript that in vitro and in vivo are in italic.

3.      Page 11, section 3.2.2. cancer inhibition part, line 479:  "and is associated with malignant biological behaviors of cancer cells such as proliferation, migration, invasion, stemness, apoptosis, and chemo-resistance" change to : "and is associated with proliferation, migration, invasion, stemness, apoptosis, and chemo-resistance of cancer cells".

4.      Also phrase "malignant biological behaviors" is not adequate by my opinion.

5.      Page 15, section 6. Conclusion, line 657 change : "Increasingly, numerous studies " in the sentence and rephrase the sentence "Increasingly, numerous studies have demonstrated the efficacy of intensive study of lipid metabolic reprogramming, especially long-chain fatty acid oxidation, which opens up potential discoveries for targeted therapies in malignant tumors".

6.      Page 15, section 6. Conclusion, line 678 change or delete the sentence: "Therefore the target aspect of specific small molecule inhibition still needs further research."

7.      Page 15, section 6. Conclusion, line 680 rephrase the sentence: "The majority of small molecule drugs exhibit limited capacity to traverse the blood-brain barrier, in addition to other distinctive characteristics, making it a great therapeutic prospect in becoming a target for specific small molecule inhibition."

Comments on the Quality of English Language

Quality of English is good.

Author Response

Comments 1: Page 3, line 96 part: CPT1 (CPT1A, CPT1B, CPT1B) has two times CPT1B change to CPT1 (CPT1A, CPT1B, CPT1C).

Response 1: Thank you for pointing out this error.We have changed the “ CPT1 (CPT1A, CPT1B, CPT1B)” into “CPT1 (CPT1A, CPT1B, CPT1C)” in line 96.

Comments 2: Check through the manuscript that in vitro and in vivo are in italic.

Response 2: Thank you for pointing out this error. We are sorry for our carelessness. We have double-checked the words “in vitro” and “in vivo” from beginning to end and have changed the erroneous form to italics in the revised manuscript. Please see the revised text on lines 401,536,575 and 581.

Comments 3: Page 11, section 3.2.2. cancer inhibition part, line 479:  "and is associated with malignant biological behaviors of cancer cells such as proliferation, migration, invasion, stemness, apoptosis, and chemo-resistance" change to : "and is associated with proliferation, migration, invasion, stemness, apoptosis, and chemo-resistance of cancer cells".

Response 3: Thank you for your advice. According to your suggestion, we have changed "and is associated with malignant biological behaviors of cancer cells such as proliferation, migration, invasion, stemness, apoptosis, and chemo-resistance" into "and is associated with proliferation, migration, invasion, stemness, apoptosis, and chemo-resistance of cancer cells" in line 479.

Comments 4: Also phrase "malignant biological behaviors" is not adequate by my opinion.

Response 4: Thank you for your advice. We decide to delete "malignant biological behaviors" in lines 215,326,364,453,471,482,499,636.

Comments 5: Page 15, section 6. Conclusion, line 657 change : "Increasingly, numerous studies " in the sentence and rephrase the sentence "Increasingly, numerous studies have demonstrated the efficacy of intensive study of lipid metabolic reprogramming, especially long-chain fatty acid oxidation, which opens up potential discoveries for targeted therapies in malignant tumors".

Response 5: Thank you for your advice. According to your suggestion, we have changed "Increasingly, numerous studies" into "Increasingly, numerous studies have demonstrated the efficacy of intensive study of lipid metabolic reprogramming, especially long-chain fatty acid oxidation, which opens up potential discoveries for targeted therapies in malignant tumors" in line 657.

Comments 6: Page 15, section 6. Conclusion, line 678 change or delete the sentence: "Therefore the target aspect of specific small molecule inhibition still needs further research."

Response 6: Thank you for your advice. According to your suggestion, we have deleted this sentence: “Therefore the target aspect of specific small molecule inhibition still needs further research.” in the revised manuscript.

Comments 7: Page 15, section 6. Conclusion, line 680 rephrase the sentence: "The majority of small molecule drugs exhibit limited capacity to traverse the blood-brain barrier, in addition to other distinctive characteristics, making it a great therapeutic prospect in becoming a target for specific small molecule inhibition."

Response 7: Thank you for your advice. According to your suggestion, We've rewritten the sentence "The majority of small molecule drugs exhibit limited capacity to traverse the blood-brain barrier, in addition to other distinctive characteristics, making it a great therapeutic prospect in becoming a target for specific small molecule inhibition." to "This makes up for the fact that most small molecule drugs cannot cross the blood-brain barrier, showing great therapeutic prospects." in line 680.

Reviewer 4 Report

Comments and Suggestions for Authors

I am grateful to the chance provided to review this manuscript. This research was conducted to discuss the role of the carnitine palmitoyltransferase family, excluding well-characterized CPT1A and CPT1B, in cancer, respecting developing novel therapeutic interventions against the CPT family in cancer. However, I do think that several major concerns must be resolved before the publication.

Comments:

1. The introduction could be more usefully extended by providing background on the role that the carnitine palmitoyltransferase family has played in cancer metabolism, and also its possible role as a target for therapy.

2. Tables or figures summarizing the key structural properties, tissue distribution, and regulatory roles of the different isoforms (CPT1A, CPT1B, CPT1C, and CPT2) would be helpful additions to the manuscript.

3. Please include in this manuscript a figure that schematically depicts the interconnection between mitochondrial metabolism, oxidative phosphorylation, and tumor development, focusing on the role of the CPT family.

4. There are grammatical errors in this manuscript; therefore, this manuscript should be proofread with care for grammatical or typographical errors.

5. The manuscript may point toward the still-limited information regarding the functional roles of the less studied CPT1C and CPT2 isoforms in cancer and provide an outlook toward further research on this aspect.

6. Structural characterizations of the CPT family proteins should be included in this manuscript.

7. The clinical implication of targeting the CPT family in cancer therapy could be discussed, referring among others to the development of drugs targeting CPT and their possible synergistic effects with other treatments against cancer.

8. Further discussion in the manuscript can be made on the possible limitation in the current understanding regarding the regulatory mechanisms and signaling pathways involving the CPT family in cancer, and also the future research directions to be taken in light of these gaps.

Comments on the Quality of English Language

There are grammatical errors in this manuscript; therefore, this manuscript should be proofread with care for grammatical or typographical errors.

Author Response

Comments 1: The introduction could be more usefully extended by providing background on the role that the carnitine palmitoyltransferase family has played in cancer metabolism, and also its possible role as a target for therapy.

Response 1:  Thank you for your comment. Accordingly, we have supplemented the background on the role that the carnitine palmitoyltransferase family has played in cancer metabolism, and also its possible role as a target for therapy in the introduction and cite 6 articles as support. Please see the revised text from lines 54 to 68.

Comments 2: Tables or figures summarizing the key structural properties, tissue distribution, and regulatory roles of the different isoforms (CPT1A, CPT1B, CPT1C, and CPT2) would be helpful additions to the manuscript.

Response 2:Thank you very much for your advice. We have a brief description of the tissue distribution and regulatory role of each member of the CPT family in lines 3 and 5 of Table 1, and we have added a summary of key tissue characteristics in line 9 of Table 1 based on your suggestion.

Comments 3: Please include in this manuscript a figure that schematically depicts the interconnection between mitochondrial metabolism, oxidative phosphorylation, and tumor development, focusing on the role of the CPT family.

Response 3: Thank you for your suggestion, based on your suggestion we have added a piece about the interlinkages between mitochondrial metabolism, oxidative phosphorylation, and tumor development of the CPT family (Please see Figure 3).

Comments 4: There are grammatical errors in this manuscript; therefore, this manuscript should be proofread with care for grammatical or typographical errors.

Response 4: Thank you for pointing out this error. And we acknowledge that there are grammar issues in the initially submitted manuscript after carefully reading it. We have corrected all these issues that we found in the resubmitted version.

Comments 5: The manuscript may point toward the still-limited information regarding the functional roles of the less studied CPT1C and CPT2 isoforms in cancer and provide an outlook toward further research on this aspect.

Response 5: Thank you for your comment. In the discussion part, we have added a summary of the role of CPT1C and CPT2 in cancer mechanism research, and the possible prospects for future research, such as the study of CPT1C, which focuses more on exploring its possible catalytic substrates to provide more directions for exploring its function, and the exploration of CPT2's possible E3 ligase, deubiquitinase, which can focus on whether there is a more refined regulatory mechanism behind the phenomenon that CPT2 acts as a tumor promoter and inhibitor in different tissue classifications of the same cancer. Please see the revised text from lines 691 to 709.

Comments 6: Structural characterizations of the CPT family proteins should be included in this manuscript.

Response 6: Thank you for your advice. We have partial descriptions of the protein structures of CPT family members in 111 to 119 and 205 to 207, and we have consulted five more articles for a more comprehensive presentation, which are summarized in rows 3 and 8 of Table 1.

Comments 7: The clinical implication of targeting the CPT family in cancer therapy could be discussed, referring among others to the development of drugs targeting CPT and their possible synergistic effects with other treatments against cancer.

Response 7: Thank you for your comment. In the discussion section, we added the need for the development of drugs that target CPT and the synergy with cancer treatment. It mainly involves the current CPT inhibitors that are mostly developed as anti-angina, with large side effects, lack of clear differentiation of the classification of CPT family members, and the combination of specific targeted inhibitors of CPT family members with current clinical drugs can increase drug sensitivity and reduce drug resistance, and has a good application prospect. Please see the revised text from lines 724 to 736.

Comments 8: Further discussion in the manuscript can be made on the possible limitation in the current understanding regarding the regulatory mechanisms and signaling pathways involving the CPT family in cancer, and also the future research directions to be taken in light of these gaps.

Response 8:Thank you for your comment. In the final summary section, we further discuss the possible limitations of the current understanding of the regulatory mechanisms and signaling pathways involved in the CPT family in cancer and the possible future findings in light of these gaps, such as the current focus on upregulation of transcriptional regulation involving the CPT family and less attention on the protein level, and recent reports showing a correlation between CPT1C and tumor aging phenotype, and the succinyltransferase activity of CPT1A, The degradation of protein ubiquitination level is affected by succinylation modification of downstream target molecules, which has important reference significance for the study of other members of the CPT family. Please see the revised text from lines 689 to 701.